# ARMA-Based Segmentation of Human Limb Motion Sequences

**DOI:** 10.3390/s21165577

**Published:** 2021-08-19

**Authors:** Feng Mei, Qian Hu, Changxuan Yang, Lingfeng Liu

**Affiliations:** School of Information Engineering, East China Jiao Tong University, Nanchang 330013, China; mei_feng3@163.com (F.M.); huqian_ecjtu@163.com (Q.H.); changxuan_y@163.com (C.Y.)

**Keywords:** MoCap, IMU, ARMA, DTW, limb motion sequence segmentation, ensemble median filtering

## Abstract

With the development of human motion capture (MoCap) equipment and motion analysis technologies, MoCap systems have been widely applied in many fields, including biomedicine, computer vision, virtual reality, etc. With the rapid increase in MoCap data collection in different scenarios and applications, effective segmentation of MoCap data is becoming a crucial issue for further human motion posture and behavior analysis, which requires both robustness and computation efficiency in the algorithm design. In this paper, we propose an unsupervised segmentation algorithm based on limb-bone partition angle body structural representation and autoregressive moving average (ARMA) model fitting. The collected MoCap data were converted into the angle sequence formed by the human limb-bone partition segment and the central spine segment. The limb angle sequences are matched by the ARMA model, and the segmentation points of the limb angle sequences are distinguished by analyzing the good of fitness of the ARMA model. A medial filtering algorithm is proposed to ensemble the segmentation results from individual limb motion sequences. A set of MoCap measurements were also conducted to evaluate the algorithm including typical body motions collected from subjects of different heights, and were labeled by manual segmentation. The proposed algorithm is compared with the principle component analysis (PCA), K-means clustering algorithm (K-means), and back propagation (BP) neural-network-based segmentation algorithms, which shows higher segmentation accuracy due to a more semantic description of human motions by limb-bone partition angles. The results highlight the efficiency and performance of the proposed algorithm, and reveals the potentials of this segmentation model on analyzing inter- and intra-motion sequence distinguishing.

## 1. Introduction

Motion capture (MoCap) is a technology that uses either optical or inertial motion (IMU) sensors on a human body to record the body motions in three-dimensional space. The body motions contain a variety of action types with different semantic information [1]. Through statistical analysis of the motion data, one can obtain the motion sequences of different action types to realize the segmentation of human motion. As the basis of MoCap data analysis, motion segmentation classifies and divides different semantic action types in motion sequences, which divides a long motion sequence into different types of short motion sequences. The motion segmentation further provides a basis for the reuse, editing, and modification of a single motion sequence [2], which becomes the basis for body motion analysis.

From the perspectives of realistic MoCap applications, available data samples are usually sparse given various motion sequence types. Furthermore, motion sequence variation of the same type can be further enlarged among the samples due to the subject’s height, age, pace, etc. This poses some critical data pre-processing and algorithm generalization challenges for both statistical-model-based and neural-network-based segmentation methods. To balance the problems between algorithm efficiency and data sample requirements, and to best explore the temporal motion features of the human body, compared with the traditional ARMA model, we combine the prediction and fitting characteristics of the ARMA model in time series with the regularity of human motion in time series. The temporal inflection points in human motion sequence are calculated, and the inflection points are identified and extracted by a fitness algorithm to achieve motion sequence segmentation. This method overcomes the limitation that the ARMA model is only suitable for short-term sequence prediction, and allows the ARMA model to perform long motion sequences segmentation.

Figure 1 describes the general structure of our proposed algorithm, which is split into five major parts.

Motion sequence downsampling is performed to compress the data given the observation that most of the motions are low frequency compared with the sampling rate.Limb bone partition angle based body structural representation is performed by calculating the angles between the limb bones partition to the central spine partition for more semantic description of motion state changes.ARMA modeling of separated limbs is performed based on the limb-bone partition angle representation and individual parameterization of each limb’s ARMA model.Determination of segmentation point is performed with a goodness-of-fit algorithm to find the point with large deviation between the fitting sequences and the measurement sequence of the ARMA model.Ensemble median filtering of segmentation result of each limb was performed to obtain the final segmentation results.

From an application perspective, according to the process of frame by frame fitting of each frame data in the motion sequence according to the ARMA model, and combined with the fitness algorithm, we calculate the fitness of each frame data. The algorithm proposed in this paper can be applied to the following three major sequences.

Segment the motion sequence of a single motion type from the complex motion sequence.When there are redundant unknown motion sequences in the target motion sequence of a single action type, the unknown motion sequence can be separated from the target motion sequence to realize the cleaning of the motion sequence.Further subdivide the motion sequence of a single motion type, realize the fragmentation of a single motion type.

### Innovation and Contribution

In this work, we aim to design improved motion sequence segmentation methods with better semantic description and more robust to motion sequence variations. The main innovations and contributions of the present study are as follows.

We propose an autoregressive moving average (ARMA)-model-based segmentation method with a limb-bone partition angle based human body structural representation model. The ARMA-model-based segmentation algorithm is capable of analyzing and segmenting motion sequences without a large number of training data, neither does it depend on the type of motion sequences. The algorithm is then considered as robust to unknown motion sequences, which largely improves the segmentation efficiency and reduces time consumption of the algorithm tuning.We combine two algorithms for limb-bone partition angle characterization and the ARMA model fitting. Given that the ARMA model is suitable for short-term prediction of motion sequence, we determine that the deviation between the predicted value and the actual value of the limb motion sequence inflection point after the ARMA model fitting becomes larger via the fitness algorithm, and this is used to calculate the segmentation points.To design and evaluate the proposed segmentation algorithm, MoCap data [3] are measured on four subjects, including one female (165 cm) and three males (170∼180 cm). The MoCap data are collected by an IMU MoCap equipment of model Perception Neuron Pro by the Noitom Inc. Block A, Putian Desheng, No. 28, Xinjiekou outer street, Xicheng District, Beijing.

The remainder of this paper is divided into six sections. Section 2 provides the related work to motion sequence segmentation algorithms. Section 3 presents the generation of limb-bone partition angle sequences. In Section 4, the ARMA modeling of limb-bone partition angle motion sequence is introduced. Section 5 presents the algorithm of constructing the segmentation function of the ARMA model of limb-bone partition angle sequences. Section 6 evaluates and compares the segmentation accuracy and computation time of the proposed algorithm, the PCA, the K-means, and the BP-net segmentation algorithms. Finally, Section 7 provides conclusions.

## 2. Related Work

Research of motion sequence segmentation can be divided into three categories. The first approach is based on statistical analysis. The work in [4] proposed the benchmark data partition principle, and the number and location of segmentation points can be determined automatically by using the piecewise polynomial model and Bayesian binding strategy. The work in [5] proposed a string-based motion type labeling algorithm, which can be used for motion compression and segmentation. The works in [6,7] constructed an unsupervised, hierarchical, bottom-up motion segmentation framework, using the hierarchical alignment clustering method to segment motion. This approach relies on statistical results and needs a large number of data samples to describe the motion sequence statistics.

The second approach is based on the analysis of geometric characteristics. In [8], the distance between each joint and the center point is calculated, and the PCA is used for motion segmentation. To obtain the segmentation points, Refs. [9,10] analyze and compare shapes in a Riemannian manifold (RM) of the human pose. This kind of segmentation algorithm only uses the low-level physical information of MoCap data, resulting in a lack of semantic information in the segmentation results.

The third approach is based on deep learning and machine learning, which, similarly to the second approach, requires large data samples for model and algorithm training. In [11], the kernel time slicing (KTC) algorithm is used to make a linear search over a sliding window, which takes the minimum time point in the objective function as the output of the segmentation point. In [12], the deep semantic information of Laban motion analysis (LMA) sequences is used in a neural network algorithm, and the motion sequences in the motion database are compared for segmentation. The study in [13] used behavior cycle data to carry out double threshold multidimensional segmentation to decompose a complex motion sequence into simple dynamic linear model sequences. The study in [14] treated the segmentation as a clustering problem, and proposed a kernel sparse subspace clustering segmentation algorithm. The work in [15] used similar information in neighborhood graphs to aggregate motion sequences into motion segments of different types. In [16], the graph cutting method is used to construct an undirected weighted graph, and a Nystrom method (NM) is used to cluster data to complete motion segmentation. The work in [17] combined a density peak clustering (DPC) algorithm and an aligned clustering analysis (ACA) algorithm. The study in [18] proposed a new model for recognizing human actions from video sequences by integrating repetitive, gated recurrent neural networks across multiple scales with shearlet-based image segmentation. The idea is to increase training robustness and improve segmentation through the use of the shearlet transform. In [19], a deep learning method is provided that extracts the articulated parts of an object from a set of 3D structures corresponding to different states of the object. The segmentation module aggregates the deformation flows into piecewise rigid motions to find the articulated parts, and is based on a recurrent part extraction network. This method can segment independent and dependent motions and operates on 3D point clouds of the object under observation. The study in [20] proposed a method that simultaneously discovers suitable deep representations, as well as clusters and temporal boundaries, with the clustering process providing supervisory cues for updating temporal boundaries and training the proposed deep learning architecture. The coordinate descent optimization method is used to segment the motion sequences. In [21], a motion recognition method for multi-joint industrial robots based on end-arm vibration and back propagation (BP) neural network is proposed. A three-axis vibration sensor is installed on the last joint of the multi-joint industrial robot to obtain the vibration signals and then segment the acquired signal according to the length of time and extract the features.

The strengths and weaknesses of three kinds of segmentation approaches in the related literature are shown in Table 1.

## 3. Model of Skeleton and Acquisition of Limb-Bone Partition Angle Sequences

During human limb motion, the limb-bone partition angle sequences are obtained according to the different semantics and postures of the motion. There are four main parts: the acquisition of motion sequences, the extraction of human motion information, the establishment of bone direction vectors, and the formation of limb-bone partition angle sequences.

### 3.1. Structural Representation of Human Body

For MoCap applications, the human skeleton is represented by three parts, as shown in Figure 2a. It consists of the upper limbs, the lower limbs, and the spine.

The motion sequence is represented by the spatial location coordinates of each joint point; therefore, the data of the rotation angle of each joint point are converted into the coordinates of the joint point. Figure 2b shows the rotation order of Euler angle in the Cartesian coordinate system Z-X-Y, where the roll angle is denoted by *r*, the yaw angle is denoted by *y*, and the pitch angle is denoted by *p*. The node rotation matrix, denoted by M, is calculated according to the rotation order, as by Equation (Equation 1) [22].
(1)$$\begin{array}{cc} M & =RPY,\\ R & =R_{z}\left(-r\right)=\left[\begin{array}{ccc} \cos r & -\sin r & 0\\ \sin r & \cos r & 0\\ 0 & 0 & 1 \end{array}\right],\\ P & =R_{x}\left(-p\right)=\left[\begin{array}{ccc} 1 & 0 & 0\\ 0 & \cos p & -\sin p\\ 0 & \sin p & \cos p \end{array}\right],\\ Y & =R_{y}\left(-y\right)=\left[\begin{array}{ccc} \cos y & 0 & \sin y\\ 0 & 1 & 0\\ -\sin y & 0 & \cos y \end{array}\right], \end{array}$$
where R is the rotation matrix of the node around the Z axis, P is the rotation matrix of the node around the X axis, and Y is the rotation matrix of the node around the Y axis. By substituting *r*, *p*, and *y* into R, P, and Y, the calculation equation of rotation matrix M is obtained.

Through the rotation matrix between the parent node and the child node in Figure 2a, the position coordinate of each joint point is obtained by Equations (Equation 2) and (Equation 3).
(2)$$\left[\begin{array}{c} x_{c}\\ y_{c}\\ z_{c} \end{array}\right]=M_{r}*\left(M_{r-1}*\left(\ldots *\left(M_{2}*\left(M_{1}*\left(\begin{array}{c} x_{0}\\ y_{0}\\ z_{0} \end{array}\right)\right)\right)\right)\right),$$
(3)$$P=P_{root}+O_{r-1}+\ldots +O_{2}+O_{1}+O_{0},$$
where $M_{r}$ is the rotation matrix of joint point, $P_{root}$ is the location of the root node, and $O_{r}$ is the position of the child node relative to the parent node. When the human body performs periodic movements such as walking and running, the human limbs will switch between bending and extending postures periodically. The limbs will then show periodic variation, and the changes between limbs will form a correlation [23]. To this point, limb partition angle is introduced to improve the semantic description of the motion sequences.

In Table 2, the motion characteristics of different bone partitions are determined by the change of the size of each included angle, using Equation (Equation 4).
(4)$$\theta =<\theta_{A},\theta_{B}>=arc\cos\left(\frac{\theta_{A}*\theta_{B}}{\left|\theta_{A}\right|\left|\theta_{B}\right|}\right),$$
where θ∈[0,180°], $\theta_{A}$ and $\theta_{B}$ are the direction vectors of the central spine partition and different limb partitions, respectively. $\left\{\theta_{1},\;\theta_{2},\ldots ,\theta_{8}\right\}$ takes the average bone partition angles of the included bone partitions to reduce the 8-dimensional limb-bone partition angles sequences into 4-dimensional vector sequences. Table 3 presents the low limbs and the upper limbs bone partition angle calculation, where $\theta_{i},\;i\in \left\{a,b,c,d\right\}$ are the limb-bone partition angles.

### 3.2. Data Availability Statement

To design and evaluate the proposed segmentation algorithm, MoCap data were measured on four subjects, including three male (170∼180 cm) and one female (165 cm). The MoCap data [3] were collected by a Perception Neuron Pro model IMU MoCap equipment by Noitom Inc. This equipment includes 17 IMU located at the reference positions in Figure 2a. Each IMU includes internal adaptive filterers and was calibrated prior to each measurement. The measurements are then considered to contain negligible noise and bias effects for the motion segmentation analysis. The sampling frequency of the measurements is configured at 100 Hz to cover the bandwidth of major joint movements of a human body. Figure 3 shows different types of motion posture in the measurement, which are walking, running, raising hands, squatting, and leg raising. The total number of measurement sequence samples is 300.

The statistics of sampling frames corresponding to motion types of different heights are shown in Table 4.

### 3.3. Data Structure of BVH Files and Data Decomposition

BVH is a common file recording format for most MoCap systems, which is also used in the measurement recording in this study. A BVH file mainly contains two sections of information. The first describes the node semantic information of the 18 main nodes of the human body as shown in Figure 2a, which start from the hip node to the root node, and nest the definitions from the root node level by level. The second part is the motion capture data to be processed, which contains the number of data frames and sampling intervals. This part of the data are recorded in the form of Euler angles that is used to decompose the angular displacement of the moving object into three rotation components. The three rotation components refer to the offset angle of the moving object relative to coordinate axes of Z-X-Y in Figure 2b. Table 5 further simplifies the notation of Table 3.

As shown in Table 5, $\theta_{i_{t}}$ is the motion sequences corresponding to the limb-bone partition segment vector pinch angles $\theta_{i}$. We transform the 54-dimensional Euler angle data of each node in the BVH file into 4-dimensional limb-bone partition segment pinch angle data. By this process, we realize the dimension reduction and categorization of motion sequence data.

### 3.4. Statistical Analysis of Data

The motion sequence measurements are first analyzed based on their statistics in order to evaluate the temporal variation of the motions among subjects of different heights. In Figure 4, the motion sequences are grouped by the types identified. Each type of movement contains 25 sequence samples with different durations. To ensure a fair comparison, the data of the same motion type are normalized over the time domain. The dynamic time warping (DTW) algorithm [24] is introduced to align two motion sequences by minimizing their Euclidean distance with an optimal path. The algorithm evaluates the statistical consistency of the measurement among different subject’s specific type of motion via Equation (Equation 5).
(5)$$DTW\left(\theta_{i_{m}},\theta_{i_{n}}\right)=\min\left(\frac{\sum_{k=1}^{K}w_{k}}{K}\right),$$
where $w_{k}=dist{\left(\theta_{i_{e}},\theta_{i_{f}}\right)}_{k}$ is the Euclidean distance of the k-th sampling point between sequences. *K* is the number of frames in the sequence, k∈(1,K). The Euclidean distance $dist(\theta_{i_{e}},\theta_{i_{f}})$ of corresponding points in $\theta_{i_{m}}$ and $\theta_{i_{n}}$ sequences is calculated, e∈(1,m), f∈(1,n), $\theta_{i_{m}}$, and $\theta_{i_{n}}$ are the motion sequences of the same motion type of two subjects, provided by Equation (Equation 6).
(6)$$\begin{array}{cc} \theta_{i_{m}} & =\{\theta_{i_{1}},\theta_{i_{2}},\ldots ,\theta_{i_{e}},\ldots ,\theta_{i_{m}}\},\\ \theta_{i_{n}} & =\{\theta_{i_{1}},\theta_{i_{2}},\ldots ,\theta_{i_{f}},\ldots ,\theta_{i_{n}}\},\\ dist(\theta_{i_{e}},\theta_{i_{f}}) & =\sqrt{\sum_{e,f=1}^{m,n}{\left(\theta_{i_{e}}-\theta_{i_{f}}\right)}^{2}}, \end{array}$$
the sequence mapping *W* of two different heights of subjects is given by Equation (Equation 7).
(7)$$\begin{array}{c} W=\{w_{1},w_{2},\ldots ,w_{k},\ldots ,w_{K}\},\;\\ \max\left(i_{m},\;i_{n}\right)\leq K\leq i_{m}+i_{n}-1,\; \end{array}$$
the minimum distance between the two motion sequences after regularization is calculated by Equation (Equation 8).
(8)$$\begin{array}{c} r\left(i_{e},i_{f}\right)=d\left(\theta_{i_{e}},\theta_{i_{f}}\right)+\min\left\{r\left(i_{e}-1,i_{f}-1\right),r\left(i_{e}-1,i_{f}\right),r\left(i_{e},i_{f}-1\right)\right\}, \end{array}$$
where $d(\theta_{i_{e}},\theta_{i_{f}})$ is the distance between the current $\theta_{i_{e}}$ and $\theta_{i_{f}}$, $\theta_{i_{E}}$ and $\theta_{i_{F}}$ are the corresponding regulated sequences under the condition of the minimum distance $r(i_{e},i_{f})$ of the two motion sequences, as given by Equation (Equation 9).
(9)$$\begin{array}{cc} \theta_{i_{E}} & =\{{\hat{\theta }}_{i_{1}},\ldots ,{\hat{\theta }}_{i_{e}},\ldots ,{\hat{\theta }}_{i_{m}}\},\;\;\;\;\\ \theta_{i_{F}} & =\{{\hat{\theta }}_{i_{1}},\ldots ,{\hat{\theta }}_{i_{f}},\ldots ,{\hat{\theta }}_{i_{n}}\},\;\;\;\;\\ r_{\theta } & =\frac{\sum_{i_{E}}\sum_{i_{F}}\left(\theta_{i_{E}i_{F}}-{\bar{\theta }}_{i_{E}i_{F}}\right)\left(\theta_{i_{E}i_{F}}-{\bar{\theta }}_{i_{E}i_{F}}\right)}{\sqrt{\underset{i_{E}}{(\sum }\sum_{i_{F}}\left({\theta_{i_{E}i_{F}}-{\bar{\theta }}_{i_{E}i_{F}})}^{2}\right)(\sum_{i_{E}}\sum_{i_{F}}\left({\theta_{i_{E}i_{F}}-{\bar{\theta }}_{i_{E}i_{F}})}^{2}\right)}},\\ {\bar{r}}_{\theta } & =\frac{\sum_{\mu =1}^{\gamma }r_{\theta_{\mu }}}{\gamma }, \end{array}$$
where $\theta_{i_{E}i_{F}}$ is the motion sequence after DTW algorithm, γ is the number of data groups of the same action type, $r_{\theta_{\mu }}$ is a different motion sequence under the same motion type.

The results of the above analysis is shown in Figure 4, the similarity of various types of movement between different heights are generally higher than 70%. It shows that the motion sequences of the same type have the same characteristic among subjects of different heights.

## 4. ARMA Modeling of Limb-Bone Partition Angle Motion Sequence

The ARMA model is an important model for studying time sequences. It consists of an autoregressive (AR) model and a moving average (MA) model. In an ARMA model, the data of a variable $Y_{t}$ at any time *t* are expressed as a linear combination of its precedent observation $Y_{t-1},Y_{t-2},\ldots ,Y_{t-p}$ and historical random disturbance $\varepsilon_{t-1},\varepsilon_{t-2},\ldots ,\varepsilon_{t-q}$. The ARMA(p,q) is shown in Equation (Equation 10) [25].
(10)$$\begin{array}{cc} Y_{t} & =AR+MA,\\ AR & =c+\beta_{1}Y_{t-1}+\beta_{1}Y_{t-2}+\ldots +\beta_{p}Y_{t-p},\\ MA & =\lambda_{1}\varepsilon_{t}+\lambda_{2}\varepsilon_{t-2}+\ldots +\lambda_{q}\varepsilon_{t-q}+c, \end{array}$$
where *p* and *q* are the order of AR and MA, respectively. $\beta_{p}$ and $\lambda_{q}$ are the calculation coefficients of AR and MA respectively. *c* is the residual part.

### 4.1. Transformation between ARMA Model and Motion Feature Model

The ARMA model is combined with the characteristics between each limb-bone partition and the central spine partition in human limb motion sequences. The ARMA model for the bone angle is expressed by Equation (Equation 11).
(11)$$\theta_{i_{t}}=\beta_{i_{0}}+\beta_{i_{1}}\theta_{i_{\left(t-1\right)}}+\beta_{i_{2}}\theta_{i_{\left(t-2\right)}}+\ldots +\beta_{i_{n}}\theta_{i_{\left(t-n\right)}}+Z_{i_{t}},$$
where $\theta_{i}$ is the data to be fitted of the limb-bone partition angles, $\beta_{i_{n}}$ is the linear approximation coefficients, and $Z_{i_{t}}$ is the residual.

### 4.2. Stationarity Test of Characteristic Sequence of Angle between Limb-Bone Partition Segments

A motion sequence, denoted as $\theta_{i}$, can be predicted by an ARMA model under the condition that the sequence is stationary over the time domain. For time sequences, stationarity is denoted as wide-sense stationary, or covariance stationary, when the expectation, variance, and autocovariance do not change over time, which is expressed in Equation (Equation 12).
$$\begin{array}{cc} E(\theta_{i_{t}}) & =\alpha_{i},\;\mathrm{Var}\left(\theta_{i_{t}}\right)=\sigma_{i}^{2}, \end{array}$$
(12)$$\begin{array}{cc} \mathrm{Cov}(\theta_{i_{t}},\theta_{i_{\left(t-j\right)}}) & =c,(j=1,2,\ldots ,t-1), \end{array}$$
where E(·), Var(·), and Cov(·,·) are the expectation, variance, and covariance operators, α, σ, and *c* are invariants at different time observations. The stationarity evaluation of a motion sequence can then become a good indicator of motion changes over time.

### 4.3. Analysis of ARMA Modeling on Limb-Bone Partition Angle Sequences

The ARMA model of bone angle sequences analyzes the correlation coefficient of the limb-bone partition angle motion sequences, which is divided into autocorrelation coefficient (ACF) and partial autocorrelation coefficient (PACF).

The ACF computes the autocorrelation $\rho_{k}$ by Equation (Equation 13).
(13)$$\rho_{k}=\frac{\gamma_{k}}{\gamma_{0}},$$
where $\gamma_{k}$ and $\gamma_{0}$ are given by Equation (Equation 14).
(14)$$\begin{array}{cc} \gamma_{k} & =cov\left[\theta_{i_{t}},\theta_{i_{\left(t-k\right)}}\right]=\frac{1}{n}\sum_{t=1}^{n-k}\left[\theta_{i_{t}}-E\left(\theta_{i_{t}}\right)\right][\theta_{i_{\left(t-k\right)}}-E\left(\theta_{i_{t}}\right)],\\ \gamma_{0} & =\frac{1}{n-1}\sum_{t=1}^{n-k}{\left[\theta_{i_{t}}-E\left(\theta_{i_{t}}\right)\right]}^{2} \end{array}$$

The PACF is another important statistical sequence of the ARMA model of limb-bone partition angle sequences, expressed by Equation (Equation 15).
(15)$$\rho_{\left(\theta_{it},\theta_{i(t-1)}\right)|\left(\theta_{i(t-1)},\ldots ,\theta_{i(t-k+1)}\right)=\frac{E[\left(\theta_{it}-E\left(\theta_{it}\right)\right)\left(\theta_{i(t-k)}-E\left(\theta_{i(t-k)}\right)\right)]}{E[\left(\theta_{i(t-k)}-E{\left(\theta_{i(t-k)})\right)}^{2}\right]}}\;\;\;\;\;,$$
where the PACF is the correlation measure of the influence of $\theta_{i(t-k)}$ on $\theta_{it}$ after eliminating the interference of k-1 random variables in the motion sequence. If the ACF and PACF are “tailed”, and gradually tend to zero after q-order and p-order, respectively, it is possible to determine that the limb-bone partition angle is fitted to the ARMA model [26]. The ARMA models of limb-bone partition angle is then denoted as ${\mathrm{ARMA}}_{i}\left(p_{i},q_{i}\right)$, given that $p_{i}$ and $q_{i}$ are the lag orders of the model.

### 4.4. Parameter Estimation of ARMA Model with Angle Feature of Each Limb-Bone Partition

We use the least-squares (LS) algorithm to estimate the parameters of the ARMA model in Equation (Equation 20). The residual part $Z_{t}$ is expressed by Equation (Equation 16); therefore, the characteristic model of the angle of each limb-bone partition by Equations (Equation 16) and (Equation 17) [26].
(16)$$\begin{array}{c} Z_{i_{t}}=\lambda_{i_{1}}\varepsilon_{i_{\left(t-1\right)}}+\lambda_{i_{2}}\varepsilon_{i_{\left(t-2\right)}}+\ldots +\lambda_{i_{q}}\varepsilon_{i_{\left(t-q\right)}}+c,\\ \;i\in \{a,b,c,d\}\;\;\;\; \end{array}$$
(17)$$\begin{array}{c} {\mathrm{ARMA}}_{i}\left(p,q\right):\theta_{i_{t}}=\beta_{i_{1}}\theta_{i_{\left(t-1\right)}}+\ldots +\beta_{i_{p}}\theta_{i_{\left(t-p\right)}}+\lambda_{i_{1}}\varepsilon_{i_{\left(t-1\right)}}+\ldots +\lambda_{i_{q}}\varepsilon_{i_{\left(t-q\right)}}+c, \end{array}$$
where ${\beta_{i}}_{p}$ is the specific parameter data of lag order *p*, $\lambda_{i}$ is the specific parameter data of lag order *q*, and ${\varepsilon_{i}}_{t}$ is the residual part.

Let n+1<j<m, when $\hat{\beta }$ takes the minimum parameter data, then $\hat{\beta }$ is called β least-square estimation, expressed by Equations (Equation 18) and (Equation 19).
(18)$$\begin{array}{c} {\hat{Z}}_{i_{j}}=\theta_{i_{j}}-\left({\hat{\beta }}_{1}\theta_{i_{\left(j-1\right)}}+{\hat{\beta }}_{2}\theta_{i_{\left(j-2\right)}}+\ldots +{\hat{\beta }}_{p}\theta_{i_{\left(j-p\right)}}\right), \end{array}$$
(19)$$\begin{array}{c} s\left(\beta \right)=\sum_{j=p-1}^{m}{\left(\theta_{i_{t}}-\beta_{i_{1}}\theta_{i_{\left(t-1\right)}}-\ldots -\beta_{i_{p}}\theta_{i_{\left(t-p\right)}}\right)}^{2} \end{array}$$

The LS estimation of $\beta_{i}$ can be obtained by Equation (Equation 20).
(20)$$y_{i}=\left[\begin{array}{c} \theta_{i_{\left(p+1\right)}}\\ \theta_{i_{\left(p+2\right)}}\\ \vdots \\ \theta_{i_{n}} \end{array}\right],\;x_{i}=\left[\begin{array}{cccc} \theta_{i_{p}} & \theta_{i_{\left(p-1\right)}} & \ldots & \theta_{i_{1}}\\ \theta_{i_{\left(p+1\right)}} & \theta_{i_{p}} & \ldots & \theta_{i_{2}}\\ \vdots & \vdots & \vdots & \vdots \\ \theta_{i_{\left(n-1\right)}} & \theta_{i_{\left(n-2\right)}} & \ldots & \theta_{i_{\left(n-p\right)}} \end{array}\right],\;\;\;\;\;\;\;\;\;\;\;\;\;\;s\left(\beta_{i}\right)=\beta_{i}^{T}x_{i}^{T}x_{i}\beta_{i}-\beta_{i}^{T}x_{i}^{T}y_{i}-y_{i}^{T}x_{i}\beta_{i}+y_{i}^{T}y_{i},\;\;\;\;\;\;\;\;\;\;\;\;\;\;\;\;\;\;\;\;\;\;\;\;\;t\in \{1,2,\ldots ,p,\ldots ,n\},\;i\in \{a,b,c,d\}\;$$

The parameters of the ARMA model are eventually estimated by Equation (Equation 21).
(21)$$\begin{array}{cc} \hat{\beta } & =\frac{x_{i}^{T}y_{i}}{x_{i}^{T}x_{i}},\\ s({\hat{\beta }}_{i}) & =y_{i}^{T}y_{i}-y_{i}^{T}x_{i}\left(x_{i}^{T}x_{i}\right)-y_{i}^{T}x_{i}^{T}\beta_{i}+y_{i}^{T}y_{i}\\ \; & =\underset{\beta }{inf}s\left({\hat{\beta }}_{i}\right), \end{array}$$
where $s({\hat{\beta }}_{i})$ is the optimal parameter of ${\hat{\beta }}_{i}$ in the ARMA model.

### 4.5. Residual Sequence Test for ARMA Model of Limb-Bones Partition Angle

The main purpose of model testing is to test the good-of-fitness of the model on approximating motion sequences. The model is tested on whether sufficient information is extracted, and on whether the residual sequences are white noise sequences or not. When the model fails the test, the residual sequence will not be a white noise sequence. Hence, the model has to be reselected until the residual sequence becomes white noise again. The LS estimation of white noise variance is given by Equation (Equation 22).
(22)$$\begin{array}{cc} {\hat{\sigma }}_{i}^{2} & =\frac{1}{n-p}s\left({\hat{\beta }}_{i}\right)\\ & =\frac{1}{n-p}\left(y_{i}^{T}y_{i}-y_{i}^{T}x_{i}{\left(x_{i}^{T}x_{i}\right)}^{-1}x_{i}^{T}y_{i}\right)\\ \; & =\frac{1}{n-p}\sum_{t=p+1}^{n}{\left(\theta_{i_{t}}-{\hat{\beta }}_{i_{1}}\theta_{i_{\left(t-1\right)}}-\ldots -{\hat{\beta }}_{i_{p}}\theta_{i_{\left(t-p\right)}}\right)}^{2}, \end{array}$$
where $E(\varepsilon_{t})=0$ and $Var\left(\varepsilon_{t}\right)=\sigma_{\varepsilon }^{2}$. We determine that the ARMA model passes the residual detection when the conditions of Equation (Equation 22) are satisfied, and the relevant information of the residual part and $Y_{t}$ extraction are maximized.

### 4.6. ARMA Model Order Selection of Limb-Bones Partition Angle Based on Particle Swarm Optimization Algorithm

The particle swarm optimization (PSO) [27] algorithm has a strong ability to avoid the local extremum and achieve a global extremum; additionally, its usage is flexible and convergence speed is fast. These characteristics are the reasons it is used here for the problem of model order selection in the ARMA models, expressed by Equations (Equation 23) and (Equation 24).
(23)$$\begin{array}{c} v_{mn}\left(k+1\right)\;=\;v_{mn}\left(k\right)+c_{1}r_{1}\left(pbest_{mn}\left(k\right)-x_{mn}\left(k\right)\right)+c_{2}r_{2}\left(gbest_{mn}\left(k\right)-x_{mn}\left(k\right)\right), \end{array}$$
(24)$$\begin{array}{c} x_{mn}\left(k+1\right)\;=\;x_{mn}\left(k\right)+v_{mn}\left(k+1\right), \end{array}$$
where *m* is the m-th particle, *n* is the velocity, and *k* is the number of iterations. $c_{1}$ and $c_{2}$ are learning factors. In general, $c_{1}$ and $c_{2}$ are between [0,4]. $r_{1}$ and $r_{2}$ are random variables subject to uniform distribution in the range of [0,1]. $pbest_{mn}\left(\cdot \right)$ is the extreme value and $gbest_{mn}\left(\cdot \right)$ is the global extreme value. $x_{mn}\left(\cdot \right)$ is the [p,q] value in ARMA(p,q) of iteration *k*. The fitness F(p,q) of the ARMA model is used as the standard to decide whether the order of the model is appropriate, as given by Equation (Equation 25).
(25)$$\begin{array}{c} F\left(p,q\right)\;=\;\sqrt{\frac{1}{U}\sum_{t=1}^{U}{\left(\theta_{i_{t}}-{\hat{\theta }}_{i_{t}}\right)}^{2}}, \end{array}$$
where *U* is the number of frames of the motion sequence. $\theta_{i_{t}}$ is the original data of limb-bone partition angle, ${\hat{\theta }}_{i_{t}}$ is the estimation data of limb-bone partition angle.

## 5. Construction of Segmentation Function for ARMA Model of Limb-Bone Partition Angle Sequence

### 5.1. Motion Sequence Data Type Selection

The motion sequences are evaluated with ARMA models in different differential orders by Equation (Equation 27). Individual limb motion sequences, i.e., the right leg, left leg, right arm, and left arm, are fitted with ARMA models in first order, second order, and third order. We compare the similarity between ARMA fitting data and the measurements. The similarity of each limb motion sequence after first-order difference and third-order difference is higher than that of the second-order difference. The average fitness of the limbs are given by Equation (Equation 27).
(26)$$\begin{array}{cc} \theta_{i_{t(H)}}^{\prime } & =diff_{x}\left(\theta_{i_{t(H)}}\right), \end{array}$$
(27)$$\begin{array}{cc} \bar{\gamma } & =\frac{1-\frac{\sum_{j=1}^{g}{\left(\theta_{i_{t(H)}}^{\prime }-{\bar{\theta }}_{i_{t(H)}}^{\prime }\right)}^{2}}{\sum_{j=1}^{g}{\left({\hat{\theta }}_{i_{t(H)}}^{\prime }-{\bar{\theta }}_{i_{t(H)}}^{\prime }\right)}^{2}}}{n},\\ x & \in \{1,2,3\},\;j\in \{1,2,\ldots ,g,\ldots ,t\}, \end{array}$$
where $\theta_{i_{t(H)}}$ is the sequence of limb-bone partition angles at different heights. $\theta_{i_{t(H)}}^{\prime }$ is the sequence of $\theta_{i_{t(H)}}$ after difference of different orders. ${\hat{\theta }}_{i_{t(H)}}^{\prime }$ is the fitted sequence of $\theta_{i_{t(H)}}^{\prime }$. $\bar{\gamma }$ is the average fitness of ARMA model.

Figure 5 compares the first-order, second-order, and third-order average fitness under the different limbs. We compare the similarity between ARMA fitting data and measurement motion sequence data. The transition point, or the segmentation point, between actions in the motion sequence is not prominent enough after the first-order difference of the motion sequences. On the other hand, the difference is large from measurement sequences, after a third-order difference of the motion sequences average fitness. This probably indicates the motion information loss of the motion sequence, which reduces the accuracy of segmentation; therefore, motion sequence data after second-order difference are selected for the ARMA modeling.

### 5.2. Selection of Segmentation Windows

The measurement sequence $\theta_{i}$ is divided into windows of equal length, and the window length is set to 100. The stationarity of $\theta_{i}$ in each sequence window is tested. If the window sequence does not pass the stationarity test, it is differentiated. We use the ARMA model to fit each limb bone angle sequences, and divide the fitted sequences into different segmentation windows. The fitting coefficient $R_{\theta }^{2}$ is used to determine whether there are segmentation points in each segmentation window and output the window with segmentation points, by Equations (Equation 29) and (Equation 30) [25].
(28)$$\bar{\theta }=\frac{1}{n}\sum_{i=v}^{w}\theta_{i},$$
(29)$$\begin{array}{cc} SST_{\theta } & =\sum_{i=v}^{w}{\left(\theta_{i}-{\bar{\theta }}_{i}\right)}^{2},\\ SSE_{\theta } & =\sum_{i=v}^{w}{\left(\theta_{i}-{\hat{\theta }}_{i}\right)}^{2},\\ R_{\theta }^{2} & =1-\frac{{SSE}_{\theta }}{{SST}_{\theta }}, \end{array}$$
where $\theta_{i}$ is the measurement sequence of the limb-bone partition angle, ${\bar{\theta }}_{i}$ is the average of the measurement sequence of the limb-bone partition angle, and ${\hat{\theta }}_{i}$ is the fitting sequence of the ARMA model. The length of the selected data segmentation window is [v,w] interval, where *v* and *w* are the upper and lower bounds of the segmentation window. *n* is the number of data in the segmentation window, i.e., n=w−v. ${SSE}_{\theta }$ is the sum of squares of the residuals. ${SST}_{\theta }$ is the sum of squares of the total deviation. Fitting coefficient $R_{\theta }^{2}$ is closer to 1, and the view of $R_{\theta }^{2}\in \left[0,1\right]$ is proportional to the fitness of the model. The fitness threshold value $R_{\theta_{min}}^{2}=0.6$ [25] is set to analyze the fitting coefficient of the motion sequence segment by segment. When the fitting coefficient of the data segment is greater than the threshold, the data segment conforms to the current model fitting. On the contrary, the segmentation points are identified. The fitness of this data segment is calculated one by one by using the fitness analysis algorithm in the next section, and the minimum fitness in this data segment is selected as the segmentation point. By this method, the whole motion sequence is divided into different types of data segments.

### 5.3. Finding Segmentation Points of ARMA Model Based on Angle Feature of Limbs Bone Partition

The key idea of segmentation is to determine whether the current fitted ARMA model is suitable to continue to describe the subsequent sequence. The change of limb motion state determines the occurrence of changing points in the motion sequence. The ARMA model describes the underlying generation mechanism and relationship of data and has accurate short-term prediction ability [23]; therefore, the prediction step size of the current model is 1. When the predicted data are significantly different from measurement data, it shows that existing models cannot describe these data well. In this paper, the fitness data of the ARMA model were analyzed and calculated by the prediction information and historical information of the ARMA model. We segment the motion sequence by observing whether there are changing points in the sequence.

The confidence interval is used to describe the range in which measurement data falls into the prediction range of model, by Equation (Equation 30).
(30)$$P({\hat{\theta }}_{t+k}-1.96\delta_{t+k}<\theta_{t+k}<{\hat{\theta }}_{t+k}+1.96\delta_{t+k})\approx 0.95,$$
where $\left\{\theta_{1},\ldots ,\theta_{m},\ldots ,\theta_{t}\right\}$ be sequence of time *t*. The ARMA model *M* is established. The measurement data at (t+k) are $\theta_{t+k}$. Predicted data based on the model *M* are ${\hat{\theta }}_{t+k}$, the standard deviation of the measurement data is expressed as $\delta_{t+k}$; therefore, *A* means that the model *M* is used to describe $\theta_{t+k}$, *B* means that the measurement data fall within its corresponding confidence interval, and *C* means that measurement data are not abnormal.

Definition [25]: when data ${\hat{\theta }}_{t+k}$ fall into the 95% prediction confidence interval of its measurement data $\theta_{t+k}$, fitness SD of model M for $\theta_{t+k}$ is conditional probability P(A|B)=1. Otherwise, fitness SD is a conditional probability $P=(A|\bar{B})$ when data ${\hat{\theta }}_{t+k}$ are not within its 0.95 prediction confidence interval, thus fitness is calculated.

According to the definition, P(B|AC) means that the confidence interval of the sequence is 0.95. $P(\bar{C}|A)$ means that $\theta_{t+k}$ is the probability of abnormal data in the sequence, which is recorded as $R_{M}^{O}$. P(A) is the probability that model *M* can be used to describe a random event, P(A)=0.5. P(B|AC) is the probability that if it conforms to model *M* and is abnormal data, then it is not in its 0.95 prediction confidence interval. According to the discussion regarding abnormal data, we know that P(B|AC)=1. P(C) is the probability that the measurement data are not abnormal data, which is recorded as $R_{A}^{N}$. P(A) is the probability that the measurement data are not abnormal data. $P(\bar{B}|\bar{C})$ be the probability of conforming to *M* model and abnormal data, which is recorded as $R_{O}$. Max and Min represent the maximum and minimum values of the data contained in model *M* after removing abnormal data, respectively, and we calculate the ratio of prediction width of $w_{M}$ and $w_{t+k}$ (expressed as max-min).

The fitness of model *M* for a single datum is calculated by Equation (Equation 31) [25].
(31)$${\mathrm{SD}}_{t+k}=\left\{\begin{array}{cc} 1, & \theta_{t+k}\in \left[{\hat{\theta }}_{t+k}\pm 1.90\sigma_{t+k}\right]\\ \frac{\left(1-0.95\right)+0.5R_{M}^{O}}{R_{O}+R_{A}^{N}\left(1-R_{O}-\frac{w_{t+k}}{w_{M}}\right)}, & else \end{array}\right.,$$
which is a probability data ${\mathrm{SD}}_{t+k}\in \left[0,1\right]$. $R_{O}$, $R_{A}^{N}$, $R_{M}^{O}$, and $w_{M}$ are constants, set as ${R_{\theta }}_{min}=0.6,\;R_{A}^{N}=0.95,\;R_{M}^{O}=0.01,\;R_{O}=0.025,w_{M}=30$, where ${R_{\theta }}_{min}$ is the fitness threshold [25]. For the analysis, $R_{M}^{O}$ is the probability of abnormal data in the model fitting sequence data, $R_{A}^{N}$ is the probability of normal data in the actual data sequence, $R_{O}$ is the probability that the data in the actual data sequence are abnormal and not in its 95% confidence interval, and $w_{M}$ is the length of set segmentation window.

### 5.4. Convergence Demonstration

We expect that the proposed algorithm will achieve fast convergence of the fitness of the ARMA model to motion sequence, and can calculate the optimal fitting model. The convergence of the algorithm is demonstrated in Figure 6. The model fitness in the figure shows a clear monotone convergence after 20 iterations, confirming the effectiveness of the proposed algorithm.

## 6. Experimental Results and Analysis

Based on the measurement description in Section 3, the proposed algorithm was evaluated and compared with other segmentation algorithms. Manual segmentation points are used as reference segmentation points to calculate the segmentation accuracy.

### 6.1. Data Downsampling

The body motions are generally much slower than the sampling rate of the MoCap data, causing redundant frames in the measurements for the analysis; therefore, a downsampling of the MoCap data may reduce the computation and accelerate the segmentation estimation without losing action information.

### 6.2. Analysis of Angle Characteristics of Limb Segments Fitted by ARMA Model

In Figure 7, Figure 8, Figure 9 and Figure 10, the bone angle characteristics and the model fitting characteristics data samples from a subjects of height 180 cm are shown for the ARMA model fitting performance. The sample shows the characteristics of the motion sequence of human limbs, which is widely observed throughout the measurements.

Figure 7a, Figure 8a, Figure 9a, and Figure 10a show that the bone partitions for the same limb have periodicity in time sequences, which is consistent to the performance motion of the subject. From the figure, we see the changing trend of the included angle in the adjacent bone segments is generally similar. The lower part of Figure 7a, Figure 8a, Figure 9a, and Figure 10a are the average angle of the included angle data of adjacent bone partition of the same limb. Consequently, the ARMA model fitting and analysis of angle data of different limb segments are simplified. From the sequence fluctuation patterns in the figures, we conclude that the fluctuation range of limb-bone partition angles for the same limb varies widely for different types of movements. The fluctuation range of limb-bone partition angles is also larger for different limbs under the same movement type. This confirms the semantic description improvement by the introduced bone partition angle representation.

Figure 7b, Figure 8b, Figure 9b, and Figure 10b show the fitting characteristic of the second-order difference of the angle of each limb-bone partition. The result shows clear deterioration of the ARMA model fitting characteristic at the around changing point. As seen in the figure, the fit of the measured and model-fitted sequences is poor in the frame segment with inflection point. This confirms the design of the segmentation result in Equation (31).

### 6.3. Segmentation Determination

The segmentation points is extracted of the sequence of the limbs bone partition angle of each limb. The median filtering is applied to obtain the final set of predicted segmentation points, by Equation (Equation 32).
(32)$$\begin{array}{cc} S_{i} & =[S_{a},\;S_{b},\;S_{c},\;S_{d}]\\ \; & =\left[\begin{array}{cccc} S_{a_{1}} & S_{b_{1}} & S_{c_{1}} & S_{d_{1}}\\ S_{a_{2}} & S_{b_{2}} & S_{c_{2}} & S_{d_{2}}\\ \vdots & \vdots & \vdots & \vdots \\ S_{a_{n}} & S_{b_{n}} & S_{c_{n}} & S_{d_{n}} \end{array}\right],\\ s & =median(S_{i}), \end{array}$$
where $S_{a}$, $S_{b}$, $S_{c}$, and $S_{d}$ are the set of segmentation points with limb-bone partition angles. $median(S_{i})$ is the median value of each row vector in $S_{i}$. s is the final set of predicted segmentation points and *n* is the number of predicted segmentation points.

### 6.4. Analysis of Average Segmentation Accuracy and Average Calculation Time

The segmentation result obtained by manual segmentation is used as the reference to evaluate the segmentation of the proposed ARMA model. The index accuracy RI is used to quantitatively measure the effectiveness of the algorithm, by Equation (Equation 33).
(33)$$\begin{array}{cc} RI & =1-ER\;\;\;\\ \; & =1-\frac{\left|s-N\right|}{N}*100\%, \end{array}$$
where ER is the error rate, *N* is the total number of frames to be segmented, and N is the total number of frames per type of motion sequence. For example, when the segmented action sequence is walking before running, N is the actual number of frames in the walking state. Figure 11 is an example of segmentation point comparison between different algorithms. The BP-net [21] segmentation algorithm is based on training the limb-bone partition angle data set of the motion sequence in this paper, and then outputting the labels corresponding to each motion of the test sequence data set. The last split point is output by identifying the switching point in the label. Set the maximum training times to 1000 times and the global minimum error to 0.0001.

In Table 6, by comparing the average calculation time of various segmentation algorithms for the sample sequence, we find that the ARMA segmentation algorithm takes the least time and the BP-net segmentation algorithm takes the longest time. The main reason is the ARMA-model-based segmentation algorithm is capable to analyze and segment motion sequences without a large number of training data, which largely improves the segmentation efficiency and reduces time consumption of the algorithm tuning. The BP-net segmentation algorithm needs to train the sample sequence set for a long time, resulting in a longer overall time.

The order selection of the ARMA model based on residual whiteness in Section 4.4 is compared with that based on particle swarm optimization (PSO) [27] in Section 4.6. We set the particle number of the PSO algorithm to 20. In Table 6, compared with the ARMA model order selection algorithm based on residual whiteness, the calculation time of the ARMA model segmentation algorithm of the ARMA model order selection based on the PSO algorithm is reduced by 78.6 s. To select the model order of the ARMA model, we have compared the fitting value of the ARMA model with the actual value. If the actual value is similar to the predicted value, it proves that the model is established correctly. ARMA-PSO algorithm makes good use of this, avoids the complex calculation of taking the residual whiteness as the model order selection, and further reduces the computational time of the ARMA model segmentation algorithm.

We used two Intel (R) Xeon (R) CPU E5-2697 v3 @ 2.60 GHz x64 processors; 64-bit operating system. The graphics card is an NVIDIA geforce RTX2080 Ti.

We compare the algorithm accuracy with the PCA dimension reduction segmentation algorithm based on joint distance sequences [8], and the K-means clustering segmentation algorithm based on machine learning [17], as shown in Table 7. The average segmentation accuracy of the PCA segmentation algorithm is 82.0% in the segmentation of motion sequences with different heights, the average segmentation accuracy of the K-means segmentation algorithm is 90.0%, the average segmentation accuracy of the BP-net model algorithm is 91.2%, and the average segmentation accuracy of the ARMA model algorithm is 91.45%. The segmentation accuracy of the ARMA model is better than the PCA segmentation algorithm and the K-means segmentation algorithm. The segmentation accuracy of the ARMA model and BP-net algorithm is similar, and slightly better than the BP-net algorithm. The main reason is that the PCA segmentation algorithm directly extracts the main components of the distance sequences of the upper and lower limbs motion sequences after dimensionality reduction, and it does not consider the mutual constraints between the limbs. The K-means clustering segmentation algorithm directly carries out similar frames for the upper and lower limbs of the human body clustering. It mainly considers the connection between frames, but does not consider the influence and connection between limb-bone partition segments. Although the average segmentation accuracy of the BP-net algorithm is high, the algorithm takes a long time. In contrast, the ARMA algorithm extracts the angle sequences of different limb-bone partition; therefore, the BVH data file is converted into the angle between each limb-bone partition and the central spine bone, which makes it more effective to cover the semantic information of each limb motion sequence. The ARMA model is used to fit and segment the angle data of each limb sequence, which better reflects the motion characteristics of each limb in different motion states, this algorithm improves the segmentation accuracy.

## 7. Conclusions

In this paper, we propose an ARMA model motion sequence segmentation algorithm based on the limb-bone partition angle representation of human body skeletal structures. The algorithm is applied to long motion sequences based on different motion states, and it is used to calculate the angle characteristics of different limb segments and a defined spine as a central bone. The algorithm combines the accurate short-term prediction ability of the ARMA model. A fitness matching algorithm to analyze the data segment by segment and then calculate the fitness of the whole data to decide whether there is a segmentation of the data. Meanwhile, the ARMA segmentation algorithm is also used for segmenting different limb movement patterns in a single motion segment. With a comparison of the ARMA-based segmentation algorithm to the PCA, K-means, and BP-net segmentation algorithms. The PCA segmentation algorithm directly extracts the main components of the distance sequences of the upper and lower-limbs motion sequences after dimensionality reduction, which does not consider the mutual constraints between the limbs. The K-means clustering segmentation algorithm directly carries out similar frames for the upper and lower limbs of the human body clustering, and does not consider the influence and connection between limb bone segments. The BP-net segmentation algorithm is based on training the limb-bone partition angle data set of the motion sequence, which has high segmentation accuracy, but takes a long time. The improvement of the algorithm in this paper was achieved by introducing more semantic limb-bone partition angle representation to describe the human motion postures, and describe the limb motion sequence in more detail; therefore, the segmentation of the algorithm is more accurate.

The segmentation rate of motion sequences with similar motion states is slightly lower than that of motion sequences with different motion styles, when the algorithm is applied in segment of similar motion sequences. The main reason is that the angle of bone joints in similar motion sequences is relatively similar, which leads to fuzzy segmentation boundaries, and the segmentation accuracy is slightly lower than that of other motion sequences. Future work may consider improving the segmentation accuracy of similar motion sequences, and further realize motion prediction based on the segmentation results.

## Figures and Tables

**Figure 1 sensors-21-05577-f001:**
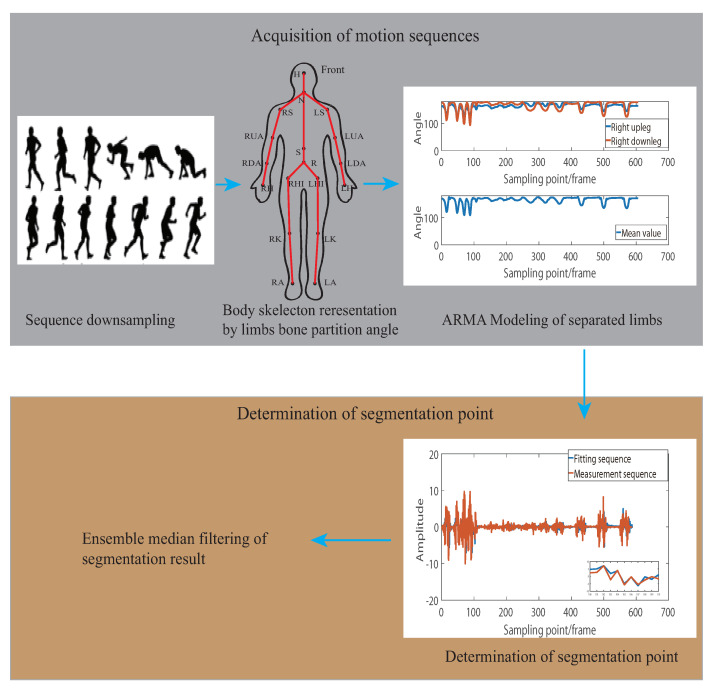
Algorithm flow and design.

**Figure 2 sensors-21-05577-f002:**
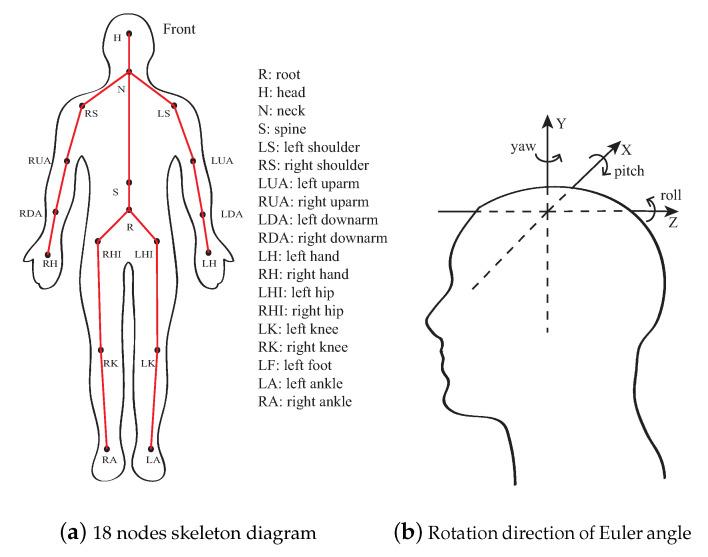
Human skeleton and Euler angle.

**Figure 3 sensors-21-05577-f003:**
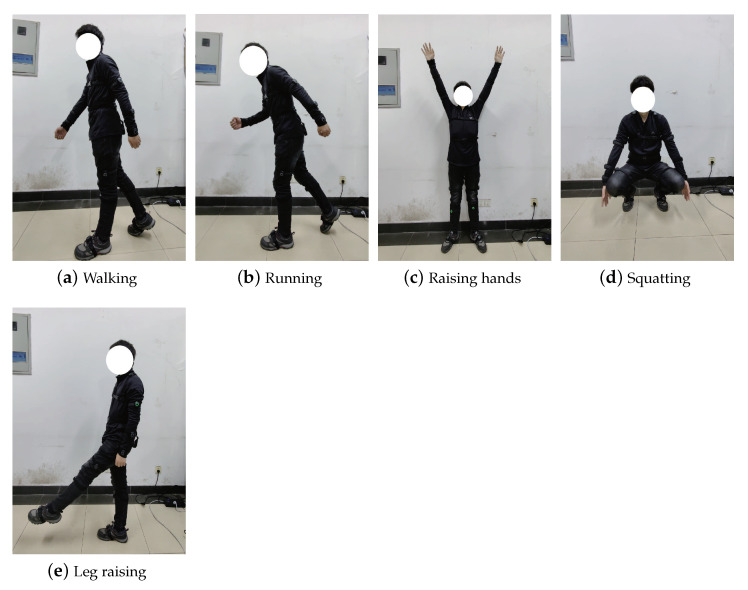
Different motion posture in motion sequences.

**Figure 4 sensors-21-05577-f004:**
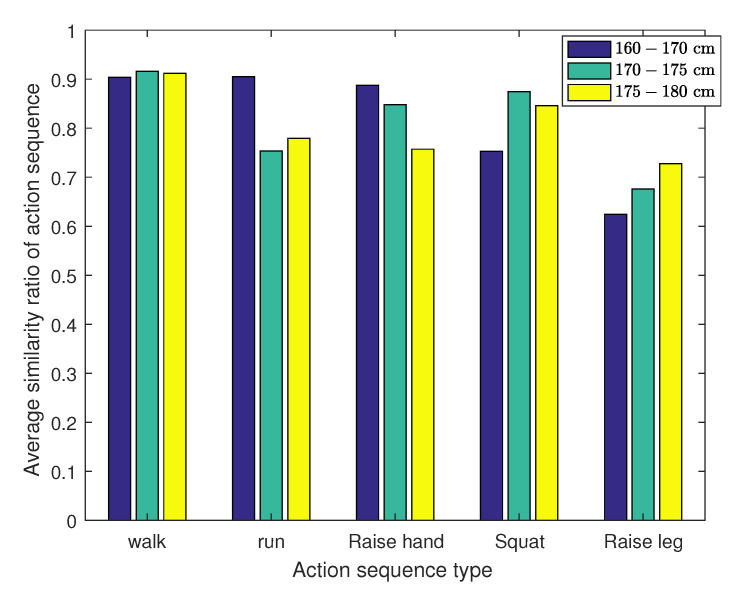
A comparative study on the consistency of different heights among the same type of movement.

**Figure 5 sensors-21-05577-f005:**
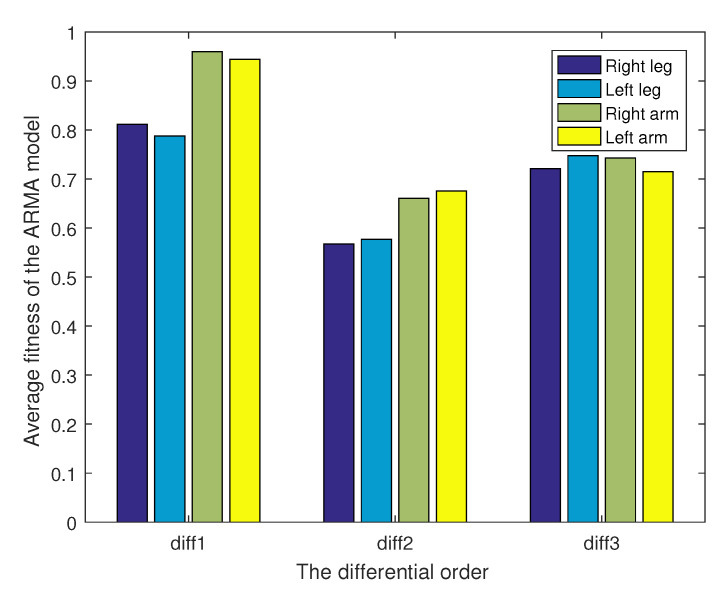
The average fitness of ARMA model data after each order difference.

**Figure 6 sensors-21-05577-f006:**
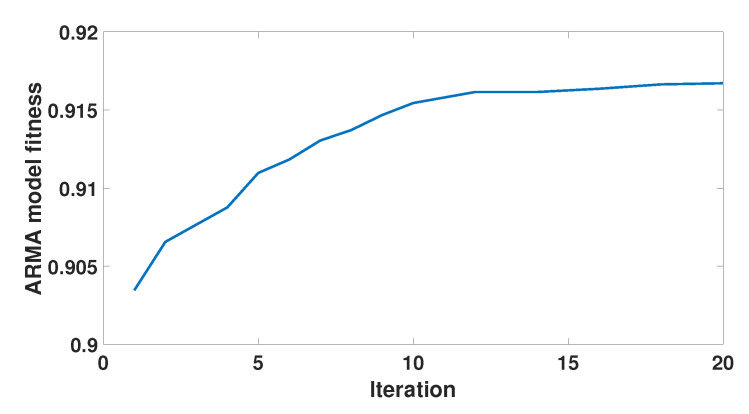
Convergence demonstration of the ARMA model segmentation algorithm.

**Figure 7 sensors-21-05577-f007:**
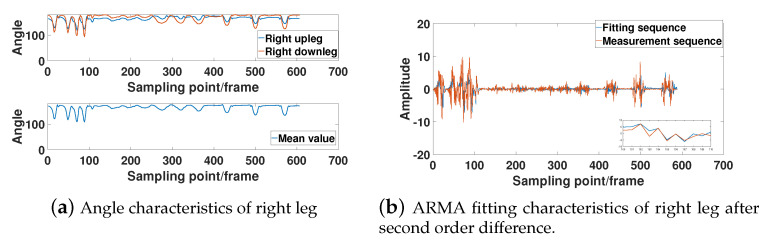
Right leg limb-bone partition angle characteristics and model fit characteristics.

**Figure 8 sensors-21-05577-f008:**
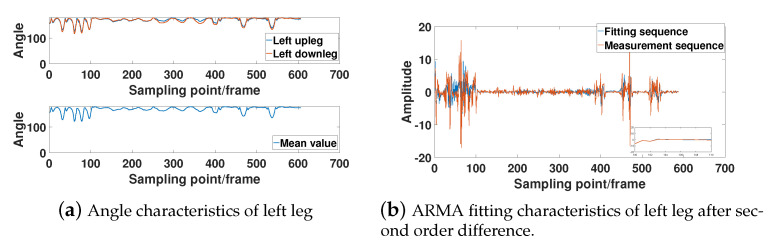
Left leg limb-bone partition angle characteristics and model fit characteristics.

**Figure 9 sensors-21-05577-f009:**
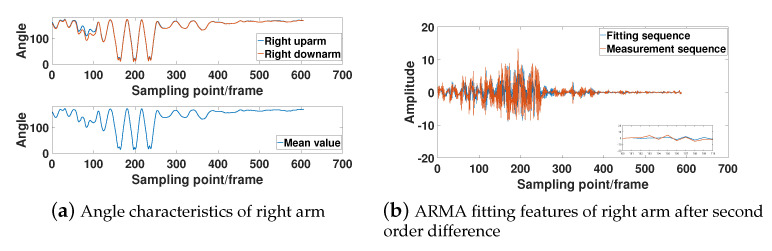
Right arm limb-bone partition angle characteristics and model fit characteristics.

**Figure 10 sensors-21-05577-f010:**
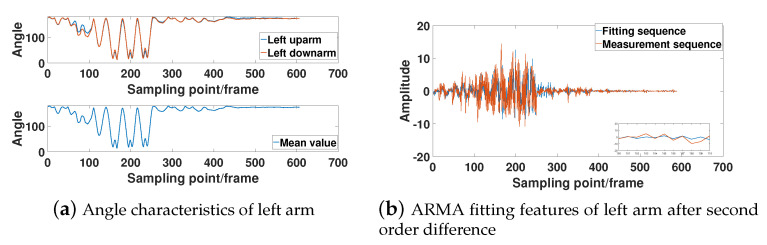
Left arm limb-bone partition angle characteristics and model fit characteristics.

**Figure 11 sensors-21-05577-f011:**
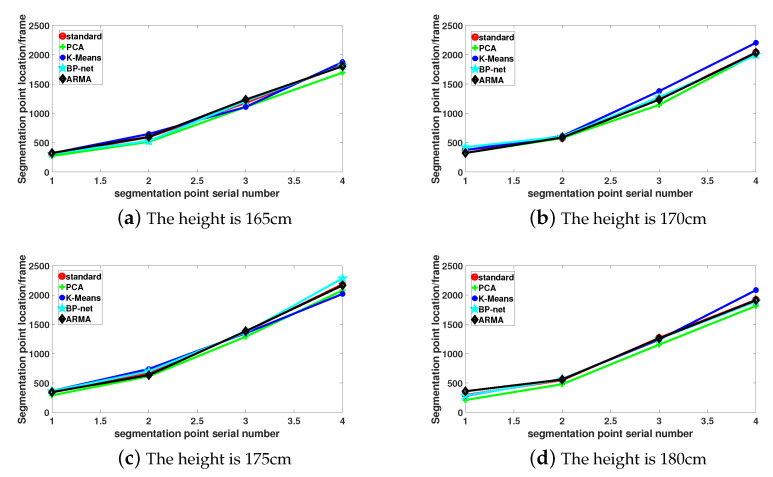
Comparison of motion sequence segmentation points of subjects with different heights.

**Table 1 sensors-21-05577-t001:** Comparison of related work on motion sequence segmentation.

Type of Segmentation Algorithm	References	Strengths	Weaknesses
Statistical characteristics	[4,5,6,7]	1. It can make full use of the data contained in the sequence. 2. The segmented sequence has strong semantics.	1. A large sample of data is needed to describe it. 2. Relying too much on statistical results.
Geometric characteristics	[8,9,10]	1. The algorithm structure is relatively simple and easy to extend.	1. The segmented sequence may lack action semantics.
Deep learning and machine learning	[11,12,13,14,15,16,17,18,19,20,21]	1. These methods can be trained to extract motion segments with high precision and speed. 2. By enhancing the quality of training samples, the semantic features of segmentation results can be improved.	1. A large number of training samples are required. 2. Such algorithms usually require a training step. The training phase highly affects the performance of these methods.

**Table 2 sensors-21-05577-t002:** Composition of limbs bone partition angle.

Low Limbs	Upper Limbs
$\theta_{1}$: RHI to RK → Central	$\theta_{5}$: RUA to RDA → Central
$\theta_{2}$: RK to RA → Central	$\theta_{6}$: RDA to RH → Central
$\theta_{3}$: LHI to LK → Central	$\theta_{7}$: LUA to LDA → Central
$\theta_{4}$: LK to LA → Central	$\theta_{8}$: LDA to LH → Central
Central bone: R → S	

**Table 3 sensors-21-05577-t003:** Simplified calculation of limbs bone partition angle.

Low Limbs	Upper Limbs
$\theta_{a}$: Right leg → Central	$\theta_{c}$: Right arm → Central
$\theta_{b}$: Left leg → Central	$\theta_{d}$: Right arm → Central
Central spine: R → S	

**Table 4 sensors-21-05577-t004:** Frame number and duration statistics of motion sequences at different heights.

Height (cm)	Number of Frames	Sequence Time Length (s)
165	2446∼2774	24.5∼27.7
170	2888∼3256	28.8∼32.6
175	2740∼2880	27.4∼28.8
180	2860∼3044	28.6∼30.4

**Table 5 sensors-21-05577-t005:** Motion sequences of limbs bone partition angle.

Bone Direction Vector	Limb-Bone Partition Angle Sequences
$\theta_{i}$	$\theta_{i_{t}}\left[\theta_{i_{1}},\;\theta_{i_{2}},\ldots ,\;\theta_{i_{n}}\right]$

**Table 6 sensors-21-05577-t006:** Comparison of the average calculation time of each segmentation algorithm.

Algorithm Type	Time (s)
ARMA-PSO	649.9
ARMA	728.5
BP-net	1646.1
PCA	742.5
K-means	1247.7

**Table 7 sensors-21-05577-t007:** Comparison of average segmentation accuracy of each algorithm.

Height (cm)	ARMA	BP-Net	PCA	K-Means
165	92.5%	91.8%	84.0%	90.4%
170	91.0%	90.5%	84.4%	88.1%
175	91.0%	91.1%	80.7%	87.1%
180	91.3%	91.4%	79.0%	90.2%

## Data Availability

All measurement data in this paper are listed in the content of the article, which can be used by all peers for related research.

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
