# Peer review of "ARMA-Based Segmentation of Human Limb Motion Sequences"

_sensors, 2021, doi:10.3390/s21165577_

Round 1

Reviewer 1 Report

In this manuscript, authors have proposed autoregressive moving average (ARMA) based segmentation of human limb motion sequences. Overall paper is well written but requires some improvements to enhance the quality of the article. 

1) Related work section should be separated from the introduction section.

2) After the related work section a comparison table is required that can summarise previous works on the motion sequence segmentation and also strengths and weaknesses of the previous and proposed method should be included in that. 

3) A separate heading should be dedicated to the novelties or your contributions section. 

4) Please check the spellings and grammar carefully as I can find some typos in the manuscript for e.g. hight>>>height in Table 5.

5) I can not find any link or details of the collected Mocap data from the 4 subjects. Please provides its link for comparison and research purposes. I am wondering the collected data from only four subjects are enough for this study? 

Reviewer 2 Report

In this paper an authoregressive method is proposed for the problem of human motion segmentation. Paper writing and organisation are sound, however, a proofread is required to improve the readability. These are my concerns and comments:

1.  What are the advantages and disadvantages of the three approaches explained in introduction?

2. More discussions about the application of this work is required.

3. The novelty, innovation and contribution of this work is not clear.

4. Apparently, ARMA has already used for motion prediction. What have you made as a new contribution which is different from existing works?

5. The review of the related works is poor. Most of the works are outdated. You should include recent works in this field and discuss their working principles. In particular, methods based on deep learning must be given and discussed.

6. Section 2 is very short and must either be extended or merged into other sections.

7. There is a weird and irrelevant paragraph at the beginning of section 4 (lines 178-181). It must be removed.

8. The convergence of the algorithm and computation time should be shown by some experiments.

9. The proposed method is compared with very old classic techniques such as PCA or K-Means. The authors must compare their work with recent state-of-the-art works so that a fair judgement in terms of the performance of the method can be obtained.

10. The work should be compared with some recent relevant deep learning techniques which exist in the literature.

Round 2

Reviewer 1 Report

Most of my comments are addressed but my last comment (No. 5) is not properly addressed.

Your provided link to the dataset is broken (unreachable). Please check it and I highly recommend that please include it in your contributions section so that the research community can get benefit from it in future research work and comparison purposes. 

Reviewer 2 Report

The authors have made good effort to improve the paper according to the reviewer's comments. I can now recommend its acceptance. A final English proofread would be required.
